# GSK-J4 Inhibition of KDM6B Histone Demethylase Blocks Adhesion of Mantle Cell Lymphoma Cells to Stromal Cells by Modulating NF-κB Signaling

**DOI:** 10.3390/cells12152010

**Published:** 2023-08-06

**Authors:** Laia Sadeghi, Anthony P. H. Wright

**Affiliations:** Division of Biomolecular and Cellular Medicine, Department of Laboratory Medicine, Karolinska Institutet, 17177 Stockholm, Sweden; anthony.wright@ki.se

**Keywords:** CCR7, GSK-J4, histone H3K27 methylation, histone H3K4 methylation, malignant B-cell adhesion, mantle cell lymphoma, NF-κB, tumor microenvironment

## Abstract

Multiple signaling pathways facilitate the survival and drug resistance of malignant B-cells by regulating their migration and adhesion to microenvironmental niches. NF-κB pathways are commonly dysregulated in mantle cell lymphoma (MCL), but the exact underlying mechanisms are not well understood. Here, using a co-culture model system, we show that the adhesion of MCL cells to stromal cells is associated with elevated levels of KDM6B histone demethylase mRNA in adherent cells. The inhibition of KDM6B activity, using either a selective inhibitor (GSK-J4) or siRNA-mediated knockdown, reduces MCL adhesion to stromal cells. We showed that KDM6B is required both for the removal of repressive chromatin marks (H3K27me3) at the promoter region of NF-κB encoding genes and for inducing the expression of NF-κB genes in adherent MCL cells. GSK-J4 reduced protein levels of the RELA NF-κB subunit and impaired its nuclear localization. We further demonstrated that some adhesion-induced target genes require both induced NF-κB and KDM6B activity for their induction (e.g., *IL-10* cytokine gene), while others require induction of NF-κB but not KDM6B (e.g., *CCR7* chemokine gene). In conclusion, KDM6B induces the NF-κB pathway at different levels in MCL, thereby facilitating MCL cell adhesion, survival, and drug resistance. KDM6B represents a novel potential therapeutic target for MCL.

## 1. Introduction

Mantle cell lymphoma (MCL) is a B-cell malignancy characterized by malignant B-cells trafficking into the lymph nodes [1]. In addition, MCL cells frequently disseminate to the bone marrow, spleen, and gastrointestinal tract [1,2,3]. MCL is a rare B-cell non-Hodgkin lymphoma (NHL) associated with the specific chromosomal translocation t(11;14)(q13;132), which involves the rearrangement of the cyclin D1 gene (*CCND1*) resulting in cyclin D1 overexpression and aberrant cell cycle progression [4,5]. The SRY-Box Transcription Factor 11 (*SOX11*) is another marker for MCL diagnosis that serves as a marker to distinguish between two distinct disease subtypes, particularly associated with an aggressive form of MCL [6]. Recently, next-generation sequencing (NGS) has allowed the identification of somatic mutations associated with MCL progression and treatment response [7]. In addition to genomic instability, the tumor microenvironment plays an important role in MCL proliferation and survival. Within microenvironmental niches, MCL cells receive growth and survival signals from stromal cells, facilitating adhesion-mediated drug resistance (CAM-DR) that counteracts effective treatment efficiency [8,9].

Migration and adhesion of malignant B-cells to stromal cells involve cytokines, cytokine receptors, and adhesion molecules regulated by the NF-κB family of transcription factors [9,10,11]. In normal B-cells, the NF-κB pathway, activated by Bruton’s tyrosine kinase (BTK) through a scaffold complex including CARD11, BCL-10, and MALT1, regulates various B-cell functions [12,13]. B-cell malignancies commonly exhibit enhanced NF-κB signaling, where aberrant activation of the pathway upregulates the expression levels of genes involved in malignant B-cell migration and adhesion to microenvironmental niches [14,15,16,17,18]. Rahal et al. demonstrated that constitutive NF-κB activity in a subgroup of MCL patients caused resistance to the BTK inhibitor ibrutinib, highlighting the importance of NF-κB in MCL therapy resistance and relapse [19].

Activation of NF-κB transcription factor subunits (RELA/P65, RELB, c-REL, p105/p50, and p100/p52) causes their translocation to the nucleus. In the nucleus, they regulate the expression of NF-κB target genes [20,21]. The NF-κB signaling pathway is divided into canonical and non-canonical pathways, where each pathway regulates the expression levels of distinct subsets of genes. However, there is potential for crosstalk between the two pathways through proteins such as NIK (NF-κB inducing kinase) and CD40 [22]. The cause of constitutive NF-κB activity in lymphomas is not fully understood, but both intrinsic and extrinsic mechanisms seem to contribute to enhanced NF-κB activity in tumor cells. In MCL cells, mutations affecting NF-κB transcription factors or upstream components such as *CARD11*, *TRAF3*, and *NIK*, as well as mutations in the *NFKBIE* negative regulator of the pathway, are associated with disturbed NF-κB activity [19,23]. Additionally, deregulation of NF-κB, even in the absence of mutations, has been identified in lymphomas as a result of tumor microenvironmental cues [24]. In MCL, overexpression of CD40L (CD40 ligand) in the tumor microenvironment mediates abnormal NF-κB activity promoting tumor cell survival [16]. Thus, there are multiple links between changes in NF-κB activity and lymphoma development, even though the mechanisms involved are not well established.

Given the role of NF-κB in promoting B-cell malignancies, several inhibitors have been developed, including Bortezomib, a proteasomal inhibitor that disturbs NF-κB inhibitor, IκB, functions. In multiple myeloma (MM) and MCL cell lines, Bortezomib treatment induces cell death [25,26,27]. However, Bortezomib interferes with the ubiquitin-proteasome pathway, causing general defects in protein degradation, resulting in severe side effects [28]. Therefore, a better understanding of the factors driving abnormal activation of NF-κB in malignant B-cells is crucial for developing more specific and effective therapeutic agents.

Epigenetic mechanisms, including DNA methylation, histone modification, and non-coding RNA, regulate gene expression. Defects in these mechanisms can lead to aberrations in signaling pathways and contribute to tumor initiation and progression [29,30,31]. Comprehensive studies have shown that epigenetic changes play an important role in malignant B-cell survival [32,33]. Histone H3 tri-methylation at lysine-27 (H3K27me3) is a repressive epigenetic mark involved in the maintenance of embryonic stem (ES) cell pluripotency and X-chromosome inactivation [34]. Lysine demethylase 6B (KDM6B), also known as JMJD3 (Jumonji domain-containing protein-3), demethylates H3K27me3 di- and trimethyl marks, activating transcription of repressed genes [35,36]. KDM6B is overexpressed in Hodgkin’s lymphomas (HLs), diffuse large B-cell lymphoma (DLBCL), acute myeloid leukemia (AML), and MM [37,38,39,40]. It plays an important role during B-cell differentiation [38,41]. In DLBCL, AML, and solid tumors (e.g., in lung and brain), the selective KDM6B inhibitor (GSK-J4) reduces tumor cell survival and enhances chemotherapy responses, suggesting the therapeutic potential of GSK-J4 in the treatment of various types of cancer [37,42].

Although our understanding of microenvironment-mediated tumor cell survival and drug resistance has improved, the role of epigenetic mechanisms, especially histone modifications, in this process remains largely unknown [43]. Here, we demonstrate that signals from the tumor microenvironment induce upregulation of NF-κB genes in MCL cells by modifying their histone modification patterns. Specifically, our results show that KDM6B is required for demethylation of the repressive H3K27me3 mark from the promoter regions of NF-κB genes, leading to their upregulated expression and deregulation of cell-adhesion-specific gene expression. Importantly, treatment of such cells with the KDM6B selective inhibitor, GSK-J4, is sufficient to inhibit the adhesion of MCL cells to stromal cells.

## 2. Materials and Methods

### 2.1. Cell Lines, Reagents, and Antibodies for Immunoblotting

The mouse stromal cell line MS-5 was purchased from DSMZ (German Collection of Microorganisms and Cell Culture GmbH). HS-5 human stromal cell line was obtained from ATCC (American Type Culture Collection, Manassas, VA, USA). The MCL cell line JeKo-1 was purchased from DSMZ, and REC-1 was kindly provided by Dr. Christian Bastard (Ronan, France). HS-5, MS-5, and MCL cell lines were maintained in DMEM (Gibco, NY, USA) or αMEM-glutamax (Gibco) or RPMI-glutamax (Gibco), respectively supplemented with 10% heat-inactivated fetal bovine serum (HI FBS, Gibco), 100 U/mL penicillin and 100 μg/mL streptomycin. Cells were cultivated at 37 °C and 5% CO_2_. Co-cultures of JeKo-1/REC-1 with MS-5 and HS-5 at a 10:1 ratio were maintained under the same conditions as for MS-5 or HS-5 cells alone. Western blot (WB) reagents, including SDS-PAGE (4–12% Tris-Glycine) gels and nitrocellulose membranes, were purchased from Life Technologies for use with the iBlot Dry-blotting system. The blocking buffer for Western blot membranes was purchased from Li-cor. Anti-Histone H3 (1:1000, ab1791, Abcam, San Francisco, CA, USA) and Anti-NF-κB P65 (1:1000 PAS-27617, Thermo Fisher Scientific) were used for Western blotting. All incubations with primary antibody for immunoblotting were conducted at 4 °C overnight. The secondary antibody was labeled with Alexa Fluor 680 or near-infrared 800 (Molecular Probs, Eugene, OR, USA) and used at 1:20,000 dilutions for 1 h at room temperature. The membrane was scanned using an Odyssey image system (Li-core), and images were analyzed using Image J software, version 1.8. PS341 (Bortezomib), GSK-J4 (SML0701), and lipopolysaccharide (L4516) were purchased from Sigma. CCR7-PE (FAB197P) was purchased from R&D Systems (Minneapolis, MN, USA).

### 2.2. Cell Fractionation for Western Blot

JeKo-1/REC-1 cells were pelleted by centrifugation and resuspended in CE, cytoplasmic extraction, buffer (cytoplasmic fraction), which was composed of 10 mM HEPES, 60 mM KCl, 1 mM EDTA, 0.075% (*v*/*v*) NP40, 1 mM DTT and 1 mM PMSF, adjusted to pH 7.6. After 5 min of incubation on ice, the cells were pelleted, and the supernatant was collected. The pellet was washed once with NE, nuclear extraction, buffer (nuclear fraction) composed of 20 mM Tris-HCl, 420 mM NaCl, 1.5 mM MgCl_2_, 0.2 mM EDTA, 1 mM PMSF, and 25% (*v*/*v*) glycerol, adjusted to pH 8.0. The pellet was then resuspended in NE buffer and incubated on ice for 5 min. The supernatant was collected by centrifugation and subjected to gel electrophoresis.

### 2.3. Cell-Cell Binding Assay and Flow Cytometry

CellTracer Carboxyfluorescein succinimidyl ester (CFSE, C34554 ThermoFisher, Waltham, MA, USA) and Far-red (C34564 ThermoFisher, Waltham, MA, USA) were used to label the JeKo-1/REC-1 and MS-5/HS-5 cells as described fully in [44]. Briefly, after staining, cells were co-cultured for 24 h. Next, unbound B-cells were removed by washing with PBS. The bound cells, together with stromal cells, were trypsinized and then subjected to flow cytometry. After data acquisition, gating was applied to exclude cell debris and dead cells using DAPI staining. Cell aggregates were also excluded during the gating process. Subsequently, gating based on CFSE and Far-red staining was performed to separate B-cells from stromal cells. The number of events within each gate was determined, and the normalized number of adherent cells was calculated as events in the CFSE gate divided by events in the Far-red gate. Flow cytometry data were obtained using a MACSQuant Analyzer 10 (Miltenyi Biotec, Bergisch Gladbach, Germany) and analyzed using FlowJo software (FlowJo version 10).

### 2.4. siRNA Transfection and Real-Time Quantitative PCR

JeKo-1 or REC-1 cells were transfected with 200 nM specific KDM6B, CCR7, IL10, and RELA (Life technology, Hewlett, NY, USA) or control siRNA (Life technology, Hewlett, NY, USA) using the Neon transfection system (ThermoFisher, Waltham, MA, USA) (1650 V, 13 ms, 2 Pulses). Total RNA from JeKo-1 or REC-1 cells was isolated using TRIzol (ThermoFisher) followed by DNaseI treatment (Fermentas, Hanover, NH, USA). A total of 1 μg of RNA was reverse-transcribed into cDNA using a high-capacity reverse transcription kit (ThermoFisher). Real-time quantitative PCR was performed using 2 μL of cDNA using 2X SYBR Green Master Mix (Biosystems, Carlsbad, CA, USA).

### 2.5. Chromatin Immunoprecipitation (ChIP) Assay

JeKo-1/REC-1 cells [2 × 10^6^] from each condition (Sep, Susp, Adh), as well as mono-cultured MS-5 cells, were cross-linked with 4% (*w*/*v*) paraformaldehyde (PFA, Sigma-Aldrich) in PBS for 10 min at room temperature. The cross-linking was quenched with 0.125 M glycine (Sigma-Aldrich). After lysing cells in lysis buffer (0.01% SDS, 1.1% Triton X-100, 1 mM EDTA, 50 mM Tris-HCl PH 8.0, and 140 mM NaCl) containing PI (Protease inhibitor, Sigma-Aldrich), DNA was sheared using a bioruptor Pico (Diagenode, Denville, NJ, USA) with 16 cycles 30 s on, 30 s off. The antibodies used in ChIP were anti-H3K27me3 (C15410195, Diagenode), anti-H3K4me3 (C1540003, Diagenode, Denville, NJ, USA), anti-Histone H3 (ab1791, Abcam, San Francisco, CA, USA) and anti-NF-κB p65 (C15310256, Diagenode, Denville, NJ, USA). A total of one-tenth of the volume in the nuclear extract was used in each ChIP assay to obtain the input DNA control. The ChIP washing process was performed as described in [45]. ChIP assays were performed in duplicate and ChIP-enriched DNA was PCR-validated for enrichment at specific gene loci relative to the input and IgG controls. Only 1% of the starting chromatin is used as input. After obtaining Ct values, the input values were adjusted to 100% using log2 (dilution factor). Then, the percentage of input is calculated as 100 × 2 ^[(Ct_ChIP) − (Ct adjusted input)] as described in [46]. Primer sequences are available in SI Appendix, Appendix A.

### 2.6. Immunofluorescence Staining and Confocal Imaging

MS-5 cells were cultured on 35 mm glass-bottom culture dishes (Greiner Bio-one) for 24 h. JeKo-1 and REC-1 cells with a 10:1 ratio were added to the pre-cultured MS-5 cells and co-cultured for 24 h. Then cells were washed twice with PBS to remove unbound JeKo-1 and REC-1 cells. Then, cells were fixed in 4% (*w*/*v*) paraformaldehyde (PFA, Sigma-Aldrich, St. Louise, MO, USA) in PBS for 15 min at room temperature. After washing with PBS, cells were blocked with 5% BSA (Sigma-Aldrich, St. Louise, MO, USA) for 30 min at room temperature. Next, cells were incubated overnight with primary antibodies, mouse anti-CD20 (MS4A1, OriGene, Rockville, MD, USA), and rabbit anti-p65 (PAS-27617, Thermo Fisher Scientific, Bohemia, NY, USA) at 4 °C. After washing with PBS, cells were incubated with secondary antibodies (goat anti-mouse fluorochrome-conjugated Alexa 555, A32727, ThermoFisher, Waltham, MA, USA) and (goat anti-rabbit fluorochrome-conjugated Alexa 488, A32731, ThermoFisher, Waltham, MA, USA) for 1 h at room temperature. After washing with PBS, the cells were mounted using fluoroshield DAPI (F6057, Sigma-Aldrich). Cell imaging was performed using a Nikon A1 confocal microscope with a 20× objective lens and images were analyzed using NIS-elements software (v.5.30.00). For 3D images, Z-stacks with a thickness of 0.97 μm composed of 15–18 images were collected.

### 2.7. Statistical Analysis

All the assays were performed with at least three biological replicates, each with at least three technical replicates, and results were reported as the sample means. The standard error of the mean (SEM) was calculated for pairwise comparisons between means. Standard deviation (SD) was calculated to assess the variation of sample data. The normal distribution of data was assessed using the Shapiro–Wilks test. Student’s *t*-test was used to evaluate differences between groups of normally distributed values; otherwise, the Wilcoxon test was used.

## 3. Results

### 3.1. Adhesion of MCL Cells to Stromal Cells Is Associated with and Requires Induced NF-κB Activity

The interaction of MCL cells, including JeKo-1 and REC1-1 cells, with stromal cells observed in microenvironments is associated by us and others with increased expression of NF-κB signature genes [15,44,47,48]. Consistently, previously published data from our group [44] showed that expression levels of NF-κB signature gene sets were elevated in adherent JeKo-1 MCL cells compared to suspension cells in the co-culture as well as mono-cultured cells, reaching the expression levels seen in mono-cultured REC-1 cells (Appendix A). Many NF-κB encoding genes tend to be highly expressed in mono-cultured REC-1 cells compared to JeKo-1. The affected genes are found in gene sets associated with the action of both canonical and non-canonical NF-κB pathways (Appendix A).

To confirm the expression profile of genes obtained with RNA-seq, the transcript levels of four DNA-binding NF-κB components (NFκB1, NFκB2, RELA, and RELB) were measured in mono-cultured and co-cultured JeKo-1 and REC-1 MCL cells using qPCR (Figure 1A). JeKo-1 and REC-1 are MCL cell lines that represent clinical features of MCL disease and that have been commonly used by us and others [8,10,44,48,49]. For JeKo-1, higher transcript levels were seen in adherent cells compared to suspension and mono-cultured cells. Similar results were also seen in REC-1 cells, but statistical significance was not reached for all of the NF-κB encoding genes, probably due to the higher relative expression levels in non-adherent REC-1 cells compared to JeKo-1 cells. The quantitative PCR (qPCR) primers used were designed to be specific for human transcripts (Appendix A), and thus little or no expression of these genes was measured in the murine MS-5 stromal cells as expected.

Elevated transcript levels of NF-κB encoding genes in adherent cells correlated with increased levels of nuclear RELA protein (Figure 1B,C). Immunofluorescent staining showed an increase in RELA nuclear localization in adherent JeKo-1 and REC-1 cells compared to mono-cultured cells. DAPI staining was used to identify cell nuclei, and antibodies against CD20 specifically identified the cell membranes of the MCL cells, thus allowing the identification of cytoplasmic and nuclear compartments in MCL cells. The increase in nuclear localization approached the level that is achieved after the treatment of mono-cultured cells with lipopolysaccharide (LPS), a known inducer of NF-κB nuclear localization (Figure 1B,C). NF-κB activation plays an important role in the adhesion of MCL cells to stromal cells because inhibition of NF-κB activity by treatment of co-cultures with either the Bortezomib (PS341) proteasome inhibitor that targets the NF-κB system or siRNA targeting RELA (Appendix A), reduces levels of MCL cell adherence to stromal cells compared to non-treated co-cultures (Figure 1D).

### 3.2. H3K27me3 and H3K4me3 Levels Are Altered at the Promoter Region of NF-κB Genes in Adherent MCL Cells

To determine whether alterations in the expression of NF-κB encoding genes in adherent MCL cells are accompanied by changes in epigenetic marks, mono- and co-cultured JeKo-1 and REC-1 cells were subjected to chromatin immunoprecipitation (ChIP) using an antibody against H3K27me3, a repressive mark, followed by qPCR analysis using primers specific for the human NF-κB gene promoters (Appendix A). The level of H3K27me3 was reduced at the promoter region of NF-κB encoding genes upon adhesion to stromal cells (Figure 2A). In addition, the H3K4me3 level, which is a mark for transcriptionally active genes, was increased in the same regions (Figure 2B). The maximal levels of H3K27me3 and H3K4me3 measured on NF-κB gene promoters are similar to levels for well-studied control genes (Figure 2A,B) *MYOD* (myogenic differentiation 1 gene) and actin [50]. There was no significant change in the histone H3 occupancy at the promoter regions of NF-κB genes, indicating that there is no significant change in nucleosome density at the promoter regions during adhesion (Appendix A). Thus, in MCL cells, the NF-κB encoding genes appear to be both de-repressed and activated at the epigenetic level during the process of adhesion to stromal cells.

### 3.3. Histone Demethylase KDM6B Is Both Induced upon and Required for Adhesion of MCL Cells to Stroma

Our previous RNA-sequencing data, comparing transcript levels in adherent JeKo-1 cells compared to suspension cells, showed an adhesion-associated increase in the levels of the KDM6B, encoding a histone H3K27me3 demethylase [48]. The RNA-sequencing results for KDM6B were confirmed using qPCR (Figure 3A). Moreover, the basal expression levels of KDM6B were significantly higher in REC-1 cells compared to JeKo-1 cells (Figure 3A), correlating well with high basal expression levels of NF-κB genes observed in REC-1 cells (Figure 1A). To determine whether KDM6B is required for the adhesion of MCL cells to stromal cells, we used a selective inhibitor, GSK-J4, to impair KDM6B enzymatic activity. The number of adhered MCL cells was measured after 24 h of co-culture in the absence or presence of different concentrations of GSK-J4 (0, 2, 4, 6, 8, 10 nM). The results showed that GSK-J4 reduced MCL cell adhesion to stromal cells in a dose-dependent fashion in both JeKo-1 and REC-1 cells (Figure 3B). Under these conditions, GSK-J4 did not detectably affect the viability of JeKo-1 and REC-1 (Appendix A). To independently test the requirement of KDM6B for the adhesion of JeKo-1 and REC-1 cells to stromal cells, we used a siRNA knockdown approach (Appendix A). The results showed inhibitory effects on the adhesion of both cell lines that were similar in magnitude to those seen with the highest dose of GSK-J4 (Figure 3C). Collectively, these results show that KDM6B is required for the adhesion of MCL cells to stromal cells and suggest that KDM6B might be involved in adhesion-related NF-κB activation in MCL cells by altering histone modifications at the promoter region of NF-κB encoding genes.

### 3.4. Inhibition of KDM6B Using GSK-J4 Decreased Nuclear RELA Levels in MCL Cells

To further evaluate the association between GSK-J4 inhibition of KDM6B, H3K27me3 levels, and NF-κB activity, we first measured the effect of increasing GSK-J4 concentration on overall levels of H3K27me3 across the genome. As shown in Appendix A, GSK-J4 treatment resulted in a dose-dependent global increase in H3K27me3 levels in both JeKo-1 and REC-1 cells. Thus, GSK-J4 causes genome-wide changes to the epigenetic programming of the genome. To understand the consequences of this change for the NF-κB pathway, we next assessed the effect of GSK-J4 on the levels of RELA in nuclear protein-enriched fractions and in the nuclei of individual cells since nuclear translocation is an important step in NF-κB activation. Figure 4A, Appendix A show a GSK-J4 dose-dependent reduction in RELA in the nuclear fractions from both JeKo-1 and REC-1 cells. Furthermore, immunofluorescence microscopy analysis shows a significant reduction in the relative level of RELA in the nuclei of both cell lines in GSK-J4 treated cells (Figure 4B). Taken together, our results suggest that the modulation of KDM6B levels using GSK-J4 led to the enrichment of repressive epigenetic marks at the promoter region of NF-κB encoding genes, leading to reduced NF-κB activity in the nucleus.

### 3.5. Adhesion Dependent Induction of NF-κB Target Gene Expression Can Be Regulated Either Directly or Indirectly by KDM6B-Mediated Heterochromatin De-Repression

NF-κB activation is associated with increased expression levels of its target genes, including genes encoding cytokines, chemokines, and their receptors. To determine whether important examples of such genes are direct targets of NF-κB as well as whether they are also directly de-repressed by H3K27me3 demethylation during MCL cell adhesion to stromal cells, we studied *CCR7* (a chemokine receptor gene) and *IL-10* (a cytokine gene), both of which are expressed at higher levels upon adherence of JeKo-1 MCL cells to stromal cells [48,51]. As expected, there was a significant increase in the expression levels of *CCR7* and *IL-10* in JeKo-1 cells upon adhesion to stromal cells, but there was little or no increase in REC-1 cells (Figure 5A). However, the depletion of *CCR7* mRNA levels using siRNA reduced the adhesion of both JeKo-1 and REC-1 MCL cells to stromal cells, indicating the functional importance of *CCR7* in both cell lines (Figure 5B and Appendix A). In contrast, depletion of *IL-10* mRNA levels did not affect the adhesion of either cell line to stromal cells, indicating that *IL-10* was not required for the adhesion of JeKo-1 and REC-1 cells (Figure 5C and Appendix A). The relatively low H3K27me3 levels on the promoter region of *CCR7* did not change upon adhesion, whereas promoter levels of RELA were enhanced in both cell lines (Figure 5D,E). In contrast, H3K27me3 levels were reduced at the promoter region of *IL-10* in adherent JeKo-1cells (in REC-1 cells, the levels are constitutively low and are scarcely, if at all, reduced in adherent cells) while promoter RELA levels were enhanced in the same regions in both cell lines (Figure 5D,E). Histone H3K4me3 levels were relatively high under all conditions for both genes in both cell lines (Appendix A). This suggests that both genes are transcriptionally primed such that the de-repression/activation observed here may represent the transformation of stalled transcription complexes to fully functional elongation complexes.

Our data suggest that the expression of the *CCR7* gene is not regulated by the demethylation of histone H3K27me3 in the promoter region. However, inhibition of KDM6B by GSK-J4 reduced cell surface levels of CCR7 in a dose-dependent manner in both JeKo-1 and REC-1 cells, indicating that *CCR7* is an indirect target of KDM6B (Appendix A). It is likely that the inhibitory effect of GSK-J4 on *CCR7* is due to its effect on the demethylation of histone H3K27me3 in the promoters of NF-κB encoding genes. Contrastingly, GSK-J4 may impact *IL10* gene expression both directly via the *IL10* gene promoter and via the promoters of NF-κB encoding genes. Furthermore, the epigenetic regulation of the *IL10* gene appears to differ in MCL cells of different origins since it is clearly seen only in the JeKo-1 cell line.

## 4. Discussion

While MCL is still regarded as incurable, impressive response rates have been seen in patients treated with targeted agents, e.g., BTK inhibitors [49,52]. Although initial treatment with BTK inhibitors offers long-term benefits to the patients, many patients develop resistance or relapse after treatment [53]. Mantle cell lymphoma cells often acquire resistance to such therapy and continue to survive, commonly caused by mutations affecting the BTK protein [53]. BTK inhibitor treatment is characterized by the effusion of malignant cells from protective microenvironments, notably lymph nodes, a process which is mirrored in vitro by inhibition of malignant cell adhesion to normal cells that characterize protective microenvironments, notably stromal cells [15,47]. 

In the current study, we explored the impact of GSK-J4, a candidate inhibitor that could be used as a complement to BTK inhibitors, on the adhesion of MCL cells to stromal cells using an in vitro cell adhesion assay. GSK-J4 is a selective inhibitor of the KDM6B histone H3K27me3 demethylase. Previous studies have shown that GSK-J4 affects tumor cell survival by influencing the tumor microenvironment [39,54]. Here we showed that suppression of histone demethylase KDM6B with either GSK-J4 or siRNA mediated knockdown effectively inhibits the adhesion of MCL cells to stromal cells. Notably, in REC-1 cells, the inhibition of BTK did not result in a reduction in REC-1 cell adhesion to stromal cells, whereas suppression of KDM6B significantly decreased REC-1 adhesion to stromal cells [44], suggesting different mechanisms of action of the different inhibitors. The inhibitory effect of GSK-J4 is likely achieved by inhibiting the ability of KDM6B to epigenetically alter the activation of the NF-κB pathway, which is required together with KDM6B for the adhesion of MCL cells to stromal cells. Consistently, our data showed that histone H3K27me3 levels are reduced in the promoter region of NF-κB encoding genes and the NF-κB target gene, *IL-10*, most notably in adherent JeKo-1 cells but also with similar trends in adherent REC-1 cells. We have previously shown that JeKo-1 and REC-1 cells differ in adhesion-related differentially expressed genes [44], and this difference could partially depend on distinct genome-wide distributions of histone H3K27me3 in JeKo-1 and REC-1 cells. These patterns are consistent with the observed trends in the promoter regions of some genes, including *NFκB1*, *NFκB2*, and *IL-10*.

Since GSK-J4 affects the overall level of histone H3K27me3, its effects are genome-wide. Therefore, we cannot exclude hypothetical mechanisms that may be important for MCL adhesion to stromal cells in addition to those studied here. The significance of the effects shown for the NF-κB pathway is, however, consistent with previous findings [54]. Further, while the JeKo-1 and REC-1 cell lines are widely regarded as good models for MCL cells and have been widely studied, there is a clear need for future studies on clinical isolates. NF-κB genes have been identified among the differentially expressed genes in global gene expression studies comparing peripheral B-cells from MCL patients and naïve B-cells, but these studies do not inform about microenvironment-specific gene expression [33,55]. *KDM6B* was not among the differentially expressed genes in LN compared to peripheral blood of MCL patients, identified by Saba et al. [15], but this study does not exclude the possibility of its differential expression in LN or other microenvironments in MCL patients.

Due to the significant role of the NF-κB pathway in tumor survival, several compounds have been used to target NF-κB activity, including Bortezomib [56]. Bortezomib has shown remarkable clinical effects on tumor progression, mainly through the deregulation of the NF-κB pathway, but it is also associated with several adverse events, including thrombocytopenia and neurotoxicity [57]. Considering these drawbacks, it would be beneficial to explore alternative therapies that target the NF-κB pathway. One potential candidate could be GSK-J4, which has been shown by us and others to reduce NF-κB activity [54]. However, studies are, of course, required to assess the effectiveness and potential toxicity of GSK-J4 in vivo. Interestingly, in vivo studies have shown that GSK-J4 treatment reduces the level of human leukemic cells engrafted in mice and that it inhibits the growth of MDA-MB-231 breast cancer cells in xenografts without inducing toxicity to the liver or kidneys [39]. Other experiments conducted on mice have demonstrated that GSK-J4 is well tolerated in vivo [58]. Thus there are good prospects for future in vivo studies of GSK-J4 in the context of MCL.

## 5. Conclusions

Our findings suggest that the tumor microenvironment regulates the transcription levels of migration and adhesion-associated genes, possibly by altering the epigenetic landscape of tumor cells. We identified KDM6B as a key epigenetic regulator of the NF-κB transcriptional network, providing a rationale for targeting KDM6B in MCL cells. GSK-J4 inhibits the adhesion of MCL cells to stromal cells in an analogous way to the effect reported for BTK inhibitors. Future studies on clinical isolates will be needed to evaluate these findings in the context of primary cells from MCL patients.

## Figures and Tables

**Figure 1 cells-12-02010-f001:**
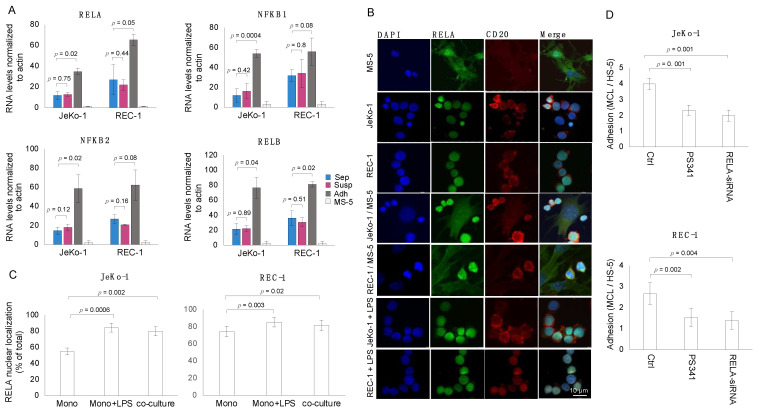
Adhesion of MCL cells to stromal cells is associated with and requires induced NF-κB activity. (**A**) qPCR data showing that the mRNA levels of four main NF-κB family members (RELA, RELB, NFκB1, and NFκB2) are increased in JeKo-1 and REC-1 cells adhered to stromal cells (Adh, dark grey) compared to suspension cells in the same co-culture (Susp, magenta) or to the cells grown separately in the absence of stromal cells (Sep, blue). The low signal observed for the murine mono-cultured MS-5 stromal cells (light grey) confirms that the primers used are specific for human sequences. Error bars show the standard deviation for three independent experiments. The *p*-values (Student’s *t*-test) indicate the significance of differences between adherent and suspension cells in the same or a separate culture as indicated. (**B**) Representative confocal microscopic images showing the localization of RELA (green) by direct immunofluorescence staining in mono-cultured cells with and without lipopolysaccharide (LPS, 50 μg/mL) stimulation and co-cultured JeKo-1 and REC-1 cells. In addition, nuclear staining (blue) and CD20 staining (red) are shown, distinguishing human B-cells from mouse stromal cells in the co-culture. (**C**) Proportion of nuclear RELA (canonical NF-κB subunit) signal intensity to total RELA signal intensity (%) in mono-cultured cells with and without LPS (50 μg/mL) stimulation and co-cultured JeKo-1 and REC-1 cells. The quantitation used Z-stack images at 0.98 μm intervals. Error bars represent the standard deviation of three independent experiments. Student’s *t*-test was performed to assess the significance of differences between LPS-stimulated or co-cultured cells in relation to unstimulated mono-cultured cells. (**D**) Dependence on NF-κB for adhesion of JeKo-1 and REC-1 cells to HS-5 stromal cells as shown using cell-cell binding assay. This was achieved by examining the effects of PS341 treatment (0.8 nM, 24 h) and RELA knockdown at a 10:1 ratio after RELA knockdown, as well as in the presence of 0.8 nM PS341 for 24 h. Student’s *t*-test was performed, and the *p*-values indicate significant differences between treated and control cells. Error bars represent the standard error of the mean (SEM).

**Figure 2 cells-12-02010-f002:**
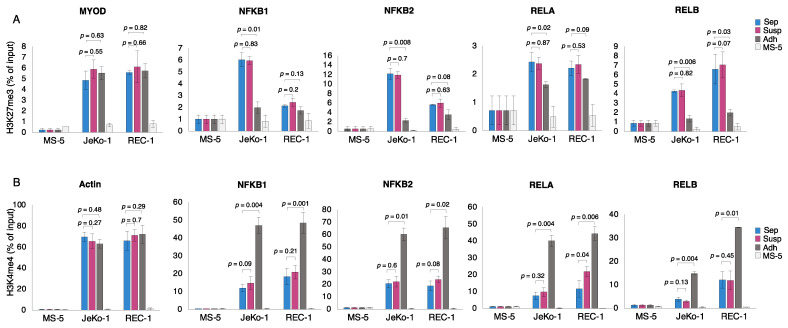
H3K27me3 and H3K4me3 levels are altered at the promoter region of NF-κB genes in adherent MCL cells. Quantitative PCR analysis of chipped DNA (ChIP-qPCR) measuring the level of (**A**) H3K27me3 and (**B**) H3K4me3 in the promoter region of NF-κB encoding genes in mono-cultured cells (Sep, blue), co-cultured suspension cells (Susp, magenta) and co-cultured adherent cells (Adh, dark gray) for JeKo-1 and REC-1 cells. Control mono-cultured MS-5 stromal cells show a low level of signal detected by the human-specific primers in the murine stromal cells (light gray). MYOD and actin promoters were used as positive control genes for H3K27me3 and H3K4me3, respectively. Data were normalized to the percent of input for each sample. Error bars show the standard deviation of three independent experiments. Student’s *t*-test was performed, and the *p*-values indicate significant differences between mono-cultured and co-cultured cells.

**Figure 3 cells-12-02010-f003:**
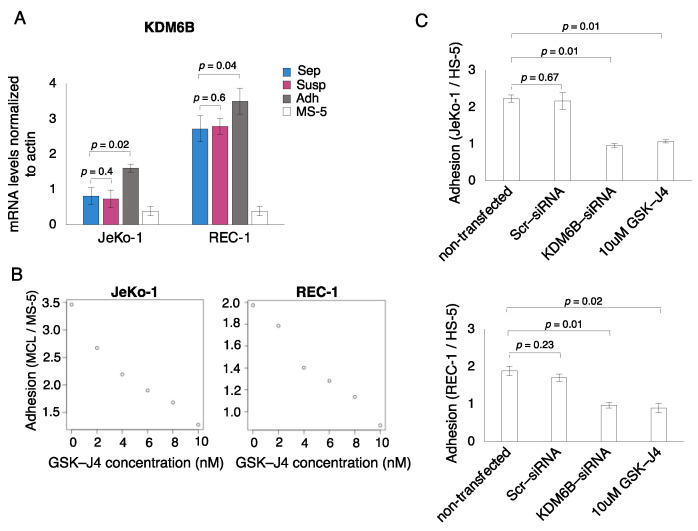
Histone demethylase KDM6B is both induced upon and required for the adhesion of MCL cells to the stroma. (**A**) qPCR measurement of KDM6B histone demethylase mRNA levels using human-specific primers showed increased levels in co-cultured adherent cells (Adh, dark gray) compared to mono-cultured cells (Sep, blue) or co-cultured suspension cells (Susp, magenta) for JeKo-1 cells and with a similar tendency for REC-1 cells. Control mono-cultured MS-5 stromal cells (MS-5, light grey) show the low level of signal detected by the human-specific primers in the murine stromal cells. Error bars show the standard deviation of three independent experiments. Student’s T-test was performed, and the *p*-values indicate the significance of differences between adherent cells and control cells. (**B**) Cell-cell binding assays were performed in the presence and absence of different concentrations of GSK-J4. Scatter plots show dose-dependent inhibition of JeKo-1 and REC-1 cell adhesion to MS-5 stromal cells by the GSK-J4 KDM6B inhibitor. (**C**) Adhesion of JeKo-1 and REC-1 cells to HS-5 cells was evaluated using cell–cell binding assays and a siRNA knockdown targeting KDM6B mRNA levels. The results demonstrate a reduced adhesion of JeKo-1 and REC-1 cells to HS-5 human stromal cells to an extent similar to treatment with GSK-J4 (10 nM). Plotted values are means from ≥3 culture wells in three independent experiments. Error bars represent the standard error of the mean (SEM). *p*-values represent statistical significance between treated and non-treated controls.

**Figure 4 cells-12-02010-f004:**
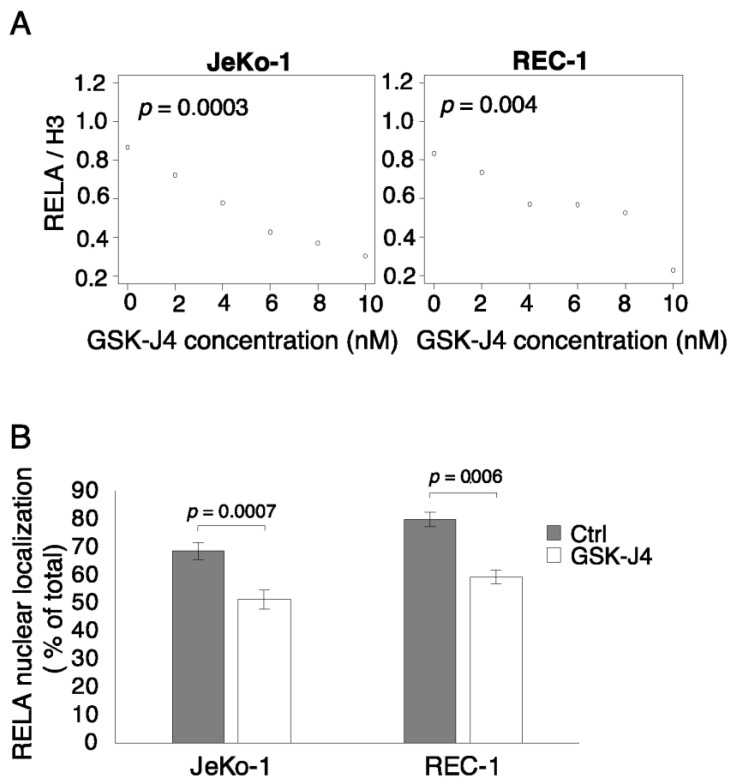
Inhibition of KDM6B using GSK-J4 decreased nuclear RELA levels in MCL cells. (**A**) Scatter plot showing dose-dependent inhibition of RELA levels by the GSK-J4 inhibitor of KDM6B in nucleus-enriched fractions of JeKo-1 and REC-1 cells using Western blot analysis. (**B**) Proportion (%) of nuclear RELA signal intensity relative to total RELA signal intensity in mono-cultured JeKo-1 and REC-1 cells in the presence and absence of GSK-J4 (10 nM). RELA immunostaining in GSK-J4 treated and untreated cells were performed, and the quantitation used Z-stack images generated at 0.98 μm intervals. Error bars represent the standard deviation of three independent experiments. Student’s *t*-test was performed, and the *p*-values indicate the significance of differences between treated and untreated cells.

**Figure 5 cells-12-02010-f005:**
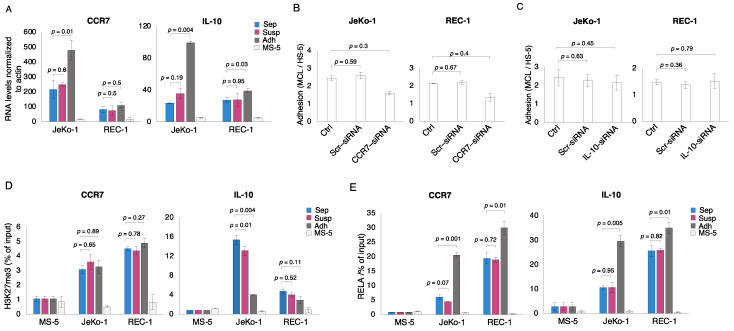
Adhesion-dependent induction of the NF-κB target gene can be regulated either directly or indirectly by KDM6B-mediated heterochromatin de-repression. (**A**) qPCR measurement of NF-κB target genes (*CCR7* and *IL-10*) using human-specific primers showed increased levels in co-cultured adherent cells (Adh, dark gray) compared to mono- (Sep, blue) or co-cultured suspension cells (Susp, magenta). Control mono-cultured MS-5 stromal cells (MS-5, light grey) show the low level of signal detected by the human-specific primers in the murine stromal cells. Error bar represents the SD of three independent experiments. Student’s *t*-test was performed, and the *p*-values indicate significant differences between mono- and co-cultured cells. (**B**) siRNA knockdown of *CCR7* mRNA levels reduces the adhesion of JeKo-1 and REC-1 cells to HS-5 human stromal cells. Error bars represent the standard error of the mean (SEM). *p*-values represent statistical significance between conditions and controls. (**C**) siRNA knockdown of *IL-10* mRNA levels do not affect the adhesion of JeKo-1 and REC-1 cells to HS-5 human stromal cells. Error bars represent the standard error of the mean (SEM). *p*-values represent statistical significance between conditions and controls. (**D**,**E**) Quantitative PCR analysis of chipped DNA (ChIP-qPCR) measuring the levels of (**D**) H3K27me3 and (**E**) RELA in the promoter region of NF-κB target genes *CCR7* and *IL-10* in mono-cultured (Sep, blue), co-cultured suspension cells (Susp, magenta) and co-cultured adherent cells (Adh, dark gray) for JeKo-1 and REC-1 cells. Control mono-cultured MS-5 stromal cells show a low level of signal detected by the human-specific primers in the murine stromal cells. Error bar represents the SD of three independent experiments. Student’s *t*-test was performed, and the *p*-values indicate significant differences between mono- and co-cultured cells.

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
