# Peer review of "GSK-J4 Inhibition of KDM6B Histone Demethylase Blocks Adhesion of Mantle Cell Lymphoma Cells to Stromal Cells by Modulating NF-κB Signaling"

_cells, 2023, doi:10.3390/cells12152010_

Round 1
Reviewer 1 Report
The authors performed extensive research to identify KDM6B as a key epigenetic regulator of the NF-κB transcriptional network, thereby providing a rationale for targeting KDM6B in MCL cells. However, some minor corrections/suggestions must be addressed to improve the manuscript. I would suggest that if it is included in the manuscript, it will be helpful for readers to carry out experiments in the future.
Major Comments:
1. Check all the “p-values” significance in figures. It has a different format in Figures 1A (NFKB1), C; Figures 4A and B
2. It would be better if authors mentioned what type of experiments were conducted to derive the result for each experiment in the result section and legends for Figures 1D; 3B, and C; 4B
3. In Figure 3C, why were the cells not treated with LPS alone or LPS with co-culture?
4. Figure 1D: Provide the Calculation for adhesion (MCL/HS-5) in the manuscript.
5. Line 117: The protocol for the cell–cell binding assay can be elaborated.
6. How is the percentage of input calculated in Figure 2?
Minor Comments:
1. Line 99: Mention cell lines in section 2.1
2.Line 101 &139: Anti-Histone H3 antibody would be preferable
3. Line 107: Remove 2x106 JeKo-1/REC-1
4. Line 107 &113: Provide abbreviation for CE and NE buffer
5. Line 129: It is 2X SYBR Green Master Mix.
6.LIne 132: The number of cells could be mentioned with a bracket in a correct format
7. Line 131: It would be better if ChIP mentioned in the heading
8. Line 140: Could be much clearer - “each ChIP”
9. How does the co-culture method Perform? This can be elaborated in detail in the supplementary section.
10. Kindly provide justification for choosing the three cell lines in the results section.
11. In NFKB1 and 2, Kappa should be italic in all places.
12. Line 211: The results of the immunofluorescence study could be explained better
13. Line 234: Abbreviation for “MYOD”
14. If possible try to follow the correct format for “p” significance (Small case and italic)
15. Follow uniformity for PS-341 and GSK-J4 (Letter ‘J’ in small case in most of the places including figures).
16. Lines 425 and 353: ‘in vivo’ and ‘via’ should be in italics.
Author Response
Thank you to all the reviewers for their thoughtful review of our manuscript. We greatly appreciate the time and effort you have dedicated to help strengthen our work. In response to your valuable feedback, we have prepared a detailed rebuttal letter that addresses each of the concerns raised point by point.
Reviewer I
- Check all the “p-values” significance in figures. It has a different format in Figures 1A (NFKB1), C; Figures 4A and B
The formatting of p-values in Figure 1 and Figure 4 was adjusted to match the format of p-values in the entire manuscript.
- It would be better if authors mentioned what type of experiments were conducted to derive the result for each experiment in the result section and legends for Figures 1D; 3B, and C; 4B
The type of experiment performed was added to Figure 1D (line 267), Figure 3B (lines 335 to 336), Figure 3C (lines 337 to 338), Figure 4A (line 367), and Figure 4B (lines 368 to 369).
- In Figure 3C, why were the cells not treated with LPS alone or LPS with co-culture?
We do not understand the basis for this comment. Figure 3C evaluates the requirement of KDM6B for adhesion of MCL cells to stromal cells. LPS is a known activator of NF-kB nuclear localization that we used as a bench mark to assess the extent of NF-kB nuclear localization induced in co-cultures (Fig 1C).
- Figure 1D: Provide the Calculation for adhesion (MCL/HS-5) in the manuscript.
Calculation was performed by dividing the number of MCL cells by the number of stromal cells in each co-culture after applying gating strategies. An elaboration of calculation was added to the Material and Methods section 2.3, lines 146 to 150
- Line 117: The protocol for the cell–cell binding assay can be elaborated.
The assay was elaborated as suggested lines 143 to 150
- How is the percentage of input calculated in Figure 2?
The percentage of input is calculated by comparing the amount of DNA in the immunoprecipitated samples to the amount of DNA in the input sample and expressing it as a percentage. The details of the calculation were added to the Materials and Methods section 2.5, specifically at lines 176 to 178
Minor Comments:
- Line 99: Mention cell lines in section 2.1
Name of cell lines were added to section 2.1 lines 108 to 112
2.Line 101 &139: Anti-Histone H3 antibody would be preferable
Corrected as suggested.
- Line 107: Remove 2x106 JeKo-1/REC-1
Corrected as suggested.
- Line 107 &113: Provide abbreviation for CE and NE buffer
Abbreviation was added to CE (line 130) and NE (line 134)
- Line 129: It is 2X SYBR Green Master Mix.
Corrected as suggested, line 161
6.LIne 132: The number of cells could be mentioned with a bracket in a correct format
Brackets were added and the format of number of cells were corrected, line 164
- Line 131: It would be better if ChIP mentioned in the heading
Corrected as suggested.
- Line 140: Could be much clearer - “each ChIP”
This has been changed to “each ChIP assay” together with some other linguistic changes in order to increase clarity (lines 172-175).
- How does the co-culture method Perform? This can be elaborated in detail in the supplementary section.
The co-culture assay has been validated previously and been fully described in reference 45 but the method is also now described in detail in Material and Methods section 2.3 lines 143 to 150
- Kindly provide justification for choosing the three cell lines in the results section.
In this study, we investigated the role of KDM6b in mantle cell lymphoma. Two well-known MCL cell lines; JeKo-1 and REC-1 were utilized. To study the role of tumor-stromal interactions, we employed human HS-5 stromal cells in co-culture experiments (Cell-cell binding assay). For expression and ChIP studies, we used mouse stromal cells, designing human-specific primers to exclude any potential mRNA and DNA from stromal cells. Detail was added to the text lines 223 to 225
- In NFKB1 and 2, Kappashould be italic in all places.
Corrected as suggested.
- Line 211: The results of the immunofluorescence study could be explained better
Some text has been added to the main text to support the more detailed explanation that is given in the figure legend, lines 236 to 238
- Line 234: Abbreviation for “MYOD”
Abbreviation was added line 283
- If possible try to follow the correct format for “p” significance (Small case and italic)
The format of all p-values is changed to small case italic in all figures
- Follow uniformity for PS-341 and GSK-J4 (Letter ‘J’ in small case in most of the places including figures).
Corrected as suggested.
- Lines 425 and 353: ‘in vivo’ and ‘via’ should be in italics.
via and in vivo format is changed to italic (via: line 407, in vivo: line 479)

Reviewer 2 Report
In the manuscript entitled “GSK-J4 inhibition of KDM6B histone demethylase blocks adhesion of Mantle Cell Lymphoma cells to stromal cells by modulating NF-kappa B signaling”, Sadeghi and Wright have studied the role of KDM6B, a histone demethylase in induction of NF-Kappa B pathway in MCL cells which plays role in adhesion, survival and drug resistance of MCL cells. They have used co-culture model to study the mechanism of adhesion of MCL cells to stromal cells. They identified the role of KDM6B in adhesion of MCL to stromal cells with elevated activity of the gene. Using a selective inhibitor (GSK-J4) or siRNA-mediated knockdown, of KDM6B, adhesion of MCL cells to stromal cells was inhibited. They further delineated the mechanism where it was demonstrated that KDM6B removes the repressive chromatin marks (H3K27me3) at the promoter region of NF-kappa B encoding genes which is required for inducing the expression of NF-kappa B regulated genes in adherent MCL cells. They have suggested KDM6B as a novel potential therapeutic target for MCL. Although the study is interesting, novel and of significance, to strengthen the data following comments can be considered:
1) The expression of KDM6B can be demonstrated in MCL patients and the benign lymphoid tissue along with the correlation with the clinic-pathological parameters.
2)The authors have used two different cell lines in in the study, it is good for understanding the mechanism, however, more cell lines can be included to demonstrate the expression level of KDM6B.
3) The authors have missed many important points in introduction, such as Mantle cell lymphoma is a non-Hodgkin lymphoma with mutation in CCND1 gene. Many other such important points are missing in the introduction part.
4) The material and method section is very unclear, although some references are provided, it is important to give brief protocol, especially about the cell lines, what was their source, what was the rationale to use them, how they were cultured, how they were isolated for experiments, what was the ratio for the culture of MCL cells with stromal cells, what was the stromal cell line used into the study and so on. All such important details are missing throughout the study.
5) Further, kindly go through the material method section and provide clarity and some details that are important for reproducibility of the data. For example, in western blotting, no details about how western blotting was done (transfer of protein, blocking, primary and secondary incubation, how the western blot was developed), similarly in all the sections of material a methods, such details are missing.
6) Statistical analysis section must be provided at the end of material method section.
7) It is also important to understand if the inhibitor GSK-J4 is inducing cell death, cell cycle arrest in MCL cells/ stromal cells.
8) Also, how these cells (MCL cells/ stromal cells) were harvested and how purity was confirmed during experiments?
Some minor suggestions:
9) Typing errors and grammatical errors needs to be carefully checked.
10) Wherever first appearing, full form must be provided and thereon abbreviation can be used. For example, KDM6B full form is not provided.
Can be improved, more clarity is needed in terms of content and language.
Author Response
Thank you to all the reviewers for their thoughtful review of our manuscript. We greatly appreciate the time and effort you have dedicated to help strengthen our work. In response to your valuable feedback, we have prepared a detailed rebuttal letter that addresses each of the concerns raised point by point.
Reviewer II
1) The expression of KDM6B can be demonstrated in MCL patients and the benign lymphoid tissue along with the correlation with the clinic-pathological parameters.
Clinical material needed for this as well as the necessary ethical permission for such studies is not available to us at this time. Thus this comment, though of potential interest, lies outside the scope of what is feasible for the present manuscript.
2)The authors have used two different cell lines in the study, it is good for understanding the mechanism, however, more cell lines can be included to demonstrate the expression level of KDM6B.
There are a few cell lines that represent the clinical features of MCL. However, in the case of MCL cell lines, most of them except JeKo-1 and REC-1 tend to bind to each other and form large clumps. As a result of this behaviour, those cell lines are not suitable for the co-culture assay as their clumping behaviour prevents reliable measurement of MCL-stromal cell interactions. For this reason, our focus has been on JeKo-1 and REC-1 cells since they don’t form clumps.
3) The authors have missed many important points in introduction, such as Mantle cell lymphoma is a non-Hodgkin lymphoma with mutation in CCND1 gene. Many other such important points are missing in the introduction part.
We thank the reviewer for insightful comment to extend the text and revise the introduction to provide a stronger and more accessible introduction to readers. In the introduction section, we have included information highlighting the role of cyclin D1 and Sox11 in mantle cell lymphoma. Moreover, we added lines regarding mutation landscape in MCL, specifically lines 30 to 39
4) The material and method section is very unclear, although some references are provided, it is important to give brief protocol, especially about the cell lines, what was their source, what was the rationale to use them, how they were cultured, how they were isolated for experiments, what was the ratio for the culture of MCL cells with stromal cells, what was the stromal cell line used into the study and so on. All such important details are missing throughout the study.
The Material and Method section was extended to include additional details regarding the cell line used, the conditions for the cell-cell binding assay, the medium composition for cell lines, and temperature were elaborated in section 2.1 lines 108 to 117. The steps and techniques employed during the cell-cell binding assay were also provided. Section 2.3 lines 143 to 150. The Western Blotting part is also elaborated in detail lines 119 to 126.
5) Further, kindly go through the material method section and provide clarity and some details that are important for reproducibility of the data. For example, in western blotting, no details about how western blotting was done (transfer of protein, blocking, primary and secondary incubation, how the western blot was developed), similarly in all the sections of material a methods, such details are missing.
The Western blotting section was elaborated in detail section 2.1, lines 119 to 126. The cell line used in the study was added, lines 108 to 116 and the cell-cell binding assay was elaborated in detail line 143 to 150.
6) Statistical analysis section must be provided at the end of material method section.
Section 2.7 was added to Material and Methods lines 198 to 204
7) It is also important to understand if the inhibitor GSK-J4 is inducing cell death, cell cycle arrest in MCL cells/ stromal cells.
In the concentrations used in this study, GSK-J4 did not cause cell death in MCL cells. However, it should be noted that we did not specifically check for cell cycle arrest. A supplementary figure (Figure S4) was added to show the effects of GSK-J4 on MCL cell death lines 315 to 316
8) Also, how these cells (MCL cells/ stromal cells) were harvested and how purity was confirmed during experiments?
MCL cell lines (JeKo-1 and REC-1) and stromal cells (HS-5 and MS-5) cell line were co-cultured and after trypsinizing we obtained a mixture of MCL and stromal cells. To separate their individual characteristics in the mixture, we either used species-specific primers (mRNA and ChIP assays), or we used differentially fluorescent labels to separate them by gating strategies in flow cytometry. In imaging experiments using confocal microscopy CD20 antibody, a cell surface receptor specific for B-cells, was used to distinguish MCL cells.
Some minor suggestions:
9) Typing errors and grammatical errors needs to be carefully checked.
Text has been carefully checked for typing and grammatical errors
10) Wherever first appearing, full form must be provided and thereon abbreviation can be used. For example, KDM6B full form is not provided.
Corrected as suggested line 85

Reviewer 3 Report
Journal of Cells
Research Article;
The article entitled “GSK-J4 inhibition of KDM6B histone demethylase blocks adhesion of Mantle Cell Lymphoma cells to stromal cells by modulating NF-kB signaling”. The authors have good efforts to investigate the Multiple signaling pathways facilitate survival and drug resistance of malignant B-cells 8 by regulating their migration and adhesion to microenvironment niches. As NF-kB pathways are commonly dysregulated in Mantle Cell Lymphoma. The inhibition of KDM6B activity, either using a selective inhibitor (GSK-J4) or siRNA-mediated knockdown, reduces MCL adhesion to stromal cells. The author showed that KDM6B is required both for removal of repressive chromatin marks (H3K27me3) at the promoter region of NF-kB encoding genes and for inducting the expression of NF-kB genes in adherent MCL cells. GSK-J4 reduced protein levels of the RELA/NF-kB subunit and impaired its nuclear localization. The author determined that some adhesion-induced target genes require both induced NF-kB and KDM6B activity for their induction while others require induction of NF-kB but not KDM6B. as KDM6B induces the NF-kB pathway at different levels in MCL thereby facilitating MCL cells adhesion, survival and drug resistance. KDM6B represents a novel potential therapeutic target for MCL.
I carefully reviewed the manuscript and found it suitable for publication in the journal. I accept this article for possible publication. There are some common mistakes in the article which should be corrected by the authors. After the correction of all the mistakes, the article could be considered for publication in the prestigious Cells Journal.
Comments for Authors
Ø Write keywords in alphabetical order and shorten the keywords.
Ø Section Introduction; The authors need to include more latest related citations in the introduction part.
Ø “Our data suggest that expression of the CCR7 genes is not regulated by demethyla- 347 tion of histone H3K27me3 in the promoter region. However, inhibition of KDM6B by 348 GSK-J4 did reduce cell surface levels of CCR7 in a dose-dependent manner in both JeKo- 349 1 and REC-1 cells indicating that CCR7 is an indirect target of KDM6B (Figure S7)” Could the author also check the effect on time dependant manner.
Ø The author needs to include an in-vivo experiment.
Ø Mentioned the original dimension clearly in Figures.
Ø Use EndNote or Mandalay software for reference sequences.
Ø Check grammatically and spelling throughout the manuscript. There are some mistakes
Cite the following references;
v https://doi: 10.2174/1871520622666220831124321
Author Response
Thank you to all the reviewers for their thoughtful review of our manuscript. We greatly appreciate the time and effort you have dedicated to help strengthen our work. In response to your valuable feedback, we have prepared a detailed rebuttal letter that addresses each of the concerns raised point by point.
Reviewer III
Comments for Authors
Ø Write keywords in alphabetical order and shorten the keywords.
Keywords have been shortened and arranged in alphabetical order.
Ø Section Introduction; The authors need to include more latest related citations in the introduction part.
Recent citations in the field of mantle cell lymphoma have been added to the manuscript References 5,7, 27, 42
Ø “Our data suggest that expression of the CCR7 genes is not regulated by demethyla- 347 tion of histone H3K27me3 in the promoter region. However, inhibition of KDM6B by 348 GSK-J4 did reduce cell surface levels of CCR7 in a dose-dependent manner in both JeKo- 349 1 and REC-1 cells indicating that CCR7 is an indirect target of KDM6B (Figure S7)” Could the author also check the effect on time dependant manner.
Answer The cell surface expression levels of CCR7 were assessed after 4 hours and 24 hours of treatment with GSK-J4 in JeKo-1 cells, and our preliminary results demonstrated that the cell surface levels of CCR7 reduced in a dose dependent manner at both time points. However, 24h treatment showed a slightly stronger effect. Since we performed all experiments after 24h treatment with GSK-J4, the dose response-curve is also conducted after 24h treatment.
Ø The author needs to include an in-vivo experiment.
There are mouse models that appear to capture important aspects of human MCL. None-the-less our choice here and elsewhere has been to focus on pathways underlying the cellular interactions between MCL cells and stromal cells. While our approach clearly has limitations, it is the best way to dissect these cell interaction related pathways. It is not practical for us to include in vivo experiments in the scope of this article, not least given the short time available for revision of the manuscript. GSK-J4 treatment has been conducted by others in mice and experiments have demonstrated no toxicity effect opening up for in vivo studies. This is elaborated in discussion section to emphasise both the limitations of the present study as well as the potential for future studies, lines 479 to 484.
Ø Mentioned the original dimension clearly in Figures.
Original dimension is fixed in Figure 1B
Ø Use EndNote or Mandalay software for reference sequences.
Endnote has been used for referencing.
Ø Check grammatically and spelling throughout the manuscript. There are some mistakes
Text was checked carefully for grammatical and spelling mistakes.
Cite the following references;
v https://doi: 10.2174/1871520622666220831124321
The suggested paper has been added as reference 27

Round 2
Reviewer 2 Report
Although the authors have addressed majority of the comments, if they included some data from patients, it could have significantly improved the weightage and quality of manuscript. However, they can at least discuss it in the conclusion part.
There is still a scope to improve the clarity of the manuscript.
Readability and clarity can be improved.
Author Response
Although the authors have addressed majority of the comments, if they included some data from patients, it could have significantly improved the weightage and quality of manuscript. However, they can at least discuss it in the conclusion part.
In the Discussion, we have added text to explain what is known from clinical global gene expression studies in MCL patients in relation to the results of this study. Lines 473 to 481
As suggested we conclude the Conclusions section by emphasising the need for corroborative studies using clinical samples.
There is still a scope to improve the clarity of the manuscript.
We have made changes that we hope further improve readability, specially in the Introduction and Discussion sections